# INFINITELY DEEP INFINITE-WIDTH NETWORKS

## ABSTRACT

Infinite-width neural networks have been extensively used to study the theoretical properties underlying the extraordinary empirical success of standard, finite-width neural networks. Nevertheless, until now, infinite-width networks have been limited to at most two hidden layers. To address this shortcoming, we study the initialisation requirements of these networks and show that the main challenge for constructing them is defining the appropriate sampling distributions for the weights. Based on these observations, we propose a principled approach to weight initialisation that correctly accounts for the functional nature of the hidden layer activations and facilitates the construction of arbitrarily many infinite-width layers, thus enabling the construction of arbitrarily deep infinite-width networks. The main idea of our approach is to iteratively reparametrise the hidden-layer activations into appropriately defined reproducing kernel Hilbert spaces and use the canonical way of constructing probability distributions over these spaces for specifying the required weight distributions in a principled way. Furthermore, we examine the practical implications of this construction for standard, finite-width networks. In particular, we derive a novel weight initialisation scheme for standard, finite-width networks that takes into account the structure of the data and information about the task at hand. We demonstrate the effectiveness of this weight initialisation approach on the MNIST, CIFAR-10 and Year Prediction MSD datasets.

## 1 INTRODUCTION

While deep neural networks have achieved impressive empirical success on many tasks across a wide range of domains in recent years (Krizhevsky et al., 2012; Simonyan & Zisserman, 2014; Szegedy et al., 2015; Sercu et al., 2015; Silver et al., 2016; Moravčík et al., 2017), they remain hard to interpret black boxes whose performance crucially depends on many heuristics. In an attempt to better understand them, significant research effort has been directed towards examining the theoretical properties of these models with one important line of research focusing on the profound connections between infinite-width networks and kernel methods. In particular, Neal (1996) established a correspondence between single-layer infinite-width networks and Gaussian processes (GP) (Rasmussen, 2006) showing the equivalence of the prior over functions that is induced by the network and a GP with a particular covariance kernel. The appropriate covariance kernel has been analytically derived for a few particular activation functions and weight priors (Williams, 1997).

Although there is a large body of research on infinite-width networks with surging recent interest in the topic (Hazan & Jaakkola, 2015; Lee et al., 2017; Matthews et al., 2018), until now the construction of these networks has been limited to at most two hidden layers. In order to overcome this shortcoming and enable deep infinite-width networks, we propose a novel approach to the construction of networks with infinitely wide hidden layers. To the best of our knowledge, this is the first method that enables the construction of infinite-width networks with more than two hidden layers. The main challenge in constructing this type of networks lies in ensuring that the inner products between the hidden layer representations and the corresponding weights, which are both functions, are well-defined. In particular, this amounts to ensuring that the weights lie in the same function space as the hidden layer representations with which the inner product is taken. In other words, the weights connecting layers $l$ and $l + 1$ need to be in the same function space as the activations of layer $l$. To construct the infinite-width layer $l + 1$, we need to construct infinitely many weights connecting layers $l$ and $l + 1$ that fulfill this requirement, i.e. we need to define a probability distribution over the function space of activations of layer $l$. As the number of layers grows, the function spaces of activations grow

increasingly more complex, thus making it increasingly more difficult to satisfy the requirements on the weights.

In order to tackle this challenge, we propose a principled approach to weight initialisation that automatically ensures that the weights are in the appropriate function space. The main idea of our approach is to make use of the canonical way of defining probability distributions over reproducing kernel Hilbert spaces (RKHS) and iteratively define the appropriate weight distributions facilitating the composition of arbitrarily many layers of infinite width. To this end, we first construct a kernel corresponding to each hidden layer and examine the associated RKHS of functions that is induced by this kernel. Next, for every layer, we establish a correspondence between the space of activations at a layer and the corresponding RKHS by reparametrising the hidden layer activations of a datapoint with the RKHS function corresponding to that point. Establishing this correspondence allows us to use a principled approach to defining probability distributions over RKHSs for constructing the appropriate sampling distribution of the weights in the infinite-width network.

We also examine some practical implications of this construction for the case of standard, finite-width neural networks in terms of weight initialisation. Using Monte Carlo approximations, we derive a novel data- and task-dependent weight initialisation scheme for finite-width networks that incorporates the structure of the data and information about the task at hand into the network.

The main contributions of this paper are

- a novel approach to the construction of infinite-width networks that enables the construction of networks with arbitrarily many hidden layers of infinite-width,

- a hierarchy of increasingly complex kernels that capture the geometry and inductive biases of individual layers in the network,

- a novel weight initialisation scheme for deep neural networks with theoretical foundations established in the infinite-width case.

The rest of the paper is organised as follows. Section 2 discusses related work and Section 3 introduces our proposed approach to the construction of deep infinite-width networks. In Section 4, we showcase the practical implications of our theoretical contribution for the case of standard, finite-width deep networks. Section 5 discusses the experimental results followed by a conclusion in Section 6.

## 2 RELATED WORK

Inspired by the groundbreaking work establishing the correspondence between single-layer neural networks and GPs (Neal, 1996; Williams, 1997), there has been a recent resurgence of interest in examining the connections between neural networks and kernel methods. In particular, a large body of literature that is concerned with constructing kernels that mimic computations in neural networks has emerged outside of the "traditional" GP-neural network correspondence context. For example, Cho & Saul (2009) construct kernels that mimic the computations in neural networks with Sign and ReLU nonlinearities, while Daniely et al. (2016) discuss the duality in expressivity between compositional kernels and neural networks, and proposes a generalised construction approach for kernels corresponding to networks whose connectivity can be represented with a directed acyclic graph. Further, Gens & Domingos (2016) construct compositional kernels that correspond to convolutional networks, while additional connections between convolutional networks and kernels are discussed in Mairal et al. (2014).

Using the multi-layer nature of neural networks within a kernel framework, Damianou & Lawrence (2013) and Duvenaud et al. (2014) discuss stacking GPs to construct deep Gaussian processes, while Wilson et al. (2016) and Al-Shedivat et al. (2017) use GPs with covariance kernels defined on the output of fully-connected and recurrent neural networks, respectively. Further examples include learning the kernel of an infinite-width neural network where the hidden layer outputs are binary (Heinemann et al., 2016), learning a neural network by kernel alignment (Duan et al., 2018) and learning in a setting when the kernel can only be estimated through sampling (Livni et al., 2017). Kernels have also been used to analyze neural networks (Montavon et al., 2011) and have been theoretically examined for consistency and universality (Steinwart et al., 2016).

There has also been a lot of recent interest in examining the "traditional" GP-neural network correspondence. In particular, Hazan & Jaakkola (2015) construct kernels for infinite-width neural networks with up to two hidden layers and arbitrary nonlinearities. In particular, a GP with a general covariance kernel is used for sampling the weights between the two hidden layers, and the hidden layer activations are assumed to be $L_2$-functions. For arbitrary network non-linearities and data distributions, the hidden layer activations do not exhibit this level of regularity. Furthermore, since the GP covariance kernel is not chosen in accordance to the structure hidden layer activations, the weights will not have the appropriate sampling distribution, and the resulting network will, in general, be ill-defined. Two recent papers (Lee et al., 2017; Matthews et al., 2018) discussed the connections between Gaussian processes and deep neural networks for increasing hidden layer width. In particular, Lee et al. (2017) derive a recurrence relation for the kernel of a deep network and use it within a GP to perform fully Bayesian inference for regression, while Matthews et al. (2018) establish theoretical results characterising the convergence in distribution of deep networks to the corresponding GP. Our work differs from these two contributions in three key aspects. First, the main motivation for our approach is enabling the construction of deep infinite-width neural networks as opposed to examining the behaviour of deep neural networks as the width of its hidden layers increases. Second, we study and discuss the initialisation requirements of infinite-width neural networks, while Lee et al. (2017) and Matthews et al. (2018) use standard initialisations in finite-width neural networks and do not properly address the changing nature of the initialisation requirements of the weights as the width of the hidden layers goes to infinity. In particular, in the limit of infinite width, the weights cannot be sampled from a normal distribution as this causes the inner product with the infinite-width hidden layer to be ill-defined. Third, as we follow different goals and approaches, the definition and utilisation of kernels differ. In particular, we define a kernel on the space of the activations of a layer, whereas Lee et al. (2017) and Matthews et al. (2018) define a kernel on the space of pre-activations of that layer. This important distinction allows us to construct the weight distributions needed for constructing deep infinite-width networks and results in the kernels encoding different aspects of the neural network behaviour.

## 3 DEEP INFINITE-WIDTH NETWORKS

In this section, we present a novel method for the construction of infinite-width networks. To the best of our knowledge, this is the first approach that enables the construction of arbitrarily deep infinite-width networks. Previous work addressing infinite-width networks was either limited to at most two layers of infinite width (Hazan & Jaakkola, 2015) or examined limits of deep finite-width networks and did not appropriately account for the changing nature of the weights in the limit of infinite width (Lee et al., 2017; Matthews et al., 2018). In particular these approaches ignore the problem that the weights connecting two adjacent infinite-width layers cannot be sampled the same way as in finite-width networks, i.e. from $\mathcal{N}(0, \sigma_w^2)$, as this does not yield a well-defined inner product with the incoming layer activations.

### 3.1 DIFFICULTY OF CONSTRUCTING NETWORKS WITH MORE THAN TWO LAYERS

In this subsection, we examine the initialisation requirements of infinite-width networks. In particular, we show that the main challenge in constructing these networks is specifying an appropriate sampling distributions for the weights that will ensure that the network as a whole is well-defined. Given an input $x \in \mathcal{X} \subseteq \mathbb{R}^d$, we compute the activation of neuron $i$ in the first layer as

$$\phi_{1,x}(w_i^0) = f_1(\langle w_i^0, x \rangle_{\mathbb{R}^d}) \tag{1}$$

with the layer non-linearity $f_1$ applied elementwise and $w_i^0$ the weights connecting the input layer to the $i$-th neuron in the first layer. Here, we make the dependence of the activations $\phi_{1,x}$ on the connecting weights explicit as it allows us to more effectively reason about the initialisation requirements. For the first layer representation $\phi_{1,x}$ to be well-defined, $\phi_{1,x}(w_i^0)$ needs to be well-defined for every neuron $i$, i.e. the inner product $\langle w_i^0, x \rangle_{\mathbb{R}^d}$ needs to be well-defined for every $i$. For the inner products to be well-defined, the weights $w_i^0$ need to lie in the same space as $x$, i.e. in $\mathbb{R}^d$. Thus, in order to construct a first infinite-width layer that is well-defined, we need to sample infinitely many weights $w_\cdot^0$ from $\mathbb{R}^d$. For this purpose, we can choose any probability measure $\nu_0$ defined on

$\mathbb{R}^d$. Let us denote by $\Phi_1$ the space of the first layer activations, i.e.

$$\Phi_1 = \overline{Span(\{\phi_{1,x} \mid x \in \mathcal{X}\})}, \quad \text{with} \quad \langle f, g \rangle_{\Phi_1} = \int f(w)g(w) d\nu_0(w) \quad \text{for} \quad f, g \in \Phi_1$$

the inner product and the bar denoting the completion of the set with respect to the norm induced by the inner product. We note that $\Phi_1$ is a space of functions as the first layer has infinite width. For the activation of neuron $i$ in the second layer, we can write

$$\phi_{2,x}(w_i^1) = f_2\big(\langle w_i^1, \phi_{1,x} \rangle_{\Phi_1}\big)$$

with the layer non-linearity $f_2$ applied elementwise and $w_i^1$ the weights connecting the first layer to the $i$-th neuron in the second layer. In order for the second layer activations $\phi_{2,x}$ to be well-defined, $\phi_{2,x}(w_i^1)$ needs to be well-defined for every $i$, i.e. the inner products $\langle w_i^1, \phi_{1,x} \rangle_{\Phi_1}$ need to be well-defined for every $i$. In order for this to be the case, the weights $w_i^1$ have to lie in the same space as $\phi_{1,x}$ for every $i$. Since $\phi_{1,x} \in \Phi_1$, we need to construct a sampling distribution over $\Phi_1$ for the weights connecting the first and second layers. As $\Phi_1$ is a space of functions defined over $\mathbb{R}^d$, a natural choice for the sampling distribution of $w_\cdot^1$ is a GP.

For constructing infinite-width networks with at most two layers, Hazan & Jaakkola (2015) endowed this GP with a general covariance function. However, as they did not take into account the structure of the underlying space $\Phi_1$, this GP will, in general, not define a sampling distribution over $\Phi_1$, thus yielding a network which is ill-defined as the weights will not sampled from the appropriate space.

Analogously to the previous two layers, in order to construct a third infinite-width layer, we need to define the weight distribution over the space of functions of functions over $\mathbb{R}^d$ that is $\Phi_2 = \overline{Span(\{\phi_{2,x} \mid x \in \mathcal{X}\})}$ as to ensure that taking inner products with $\phi_{2,x}$ is well-defined. Unfortunately, until now it was not clear how to construct this sampling distribution.

In summary, the main challenge in constructing infinite-width networks is defining appropriate sampling distributions for the weights over increasingly more complex function spaces that are induced by the hidden layer activations to ensure that the network is well-defined. In particular, in order to construct layer $l + 1$ of infinite-width, we need to construct a distribution over the function space of activations at layer $l$, thus making the inner products between the weights and the $l$-th layer activations well-defined. Constructing the appropriate weight distributions is a highly non-trivial undertaking for networks with more than two layers of infinite-width as it is not clear how to go about constructing distributions over spaces of functions of functions.

## 3.2 Constructing Deep Infinite-Width Networks

To address the above difficulty in constructing the weight distributions in infinite-width networks, we propose a principled approach to weight initialisation that automatically ensures that the weights are drawn from the appropriate function spaces. This initialisation scheme allows us to overcome the current limitation of just two hidden layers and construct deep infinite-width networks with arbitrarily many hidden layers. The main idea of our approach is to use the canonical way of defining probability distributions over RKHSs for constructing appropriate sampling distributions of the weights. To this end, we construct at every hidden layer an associated kernel and *reparametrise* the hidden layer activations into the RKHSs that are induced by these kernels. As a consequence, this yields a hierarchy of increasingly more complex kernels capturing the geometry and inductive biases of the network.

As before, let $x \in \mathcal{X} \subseteq \mathbb{R}^d$ be an input and $\phi_{1,x}(w_0^i) = f_1(\langle w_0^i, x \rangle_{\mathbb{R}^d})$ the activation of neuron $i$ in the first layer. Associated with this layer, we *define* a kernel by taking the inner product between the activations. Specifically, we define

$$k_1(x, x') = \langle \phi_{1,x}, \phi_{1,x'} \rangle_{\Phi_1} = \int \phi_{1,x}(w) \phi_{1,x'}(w) \, d\nu_0(w) \tag{2}$$

and suppress the dependence of $k_1$ on the non-linearity $f_1$ when it does not lead to notational ambiguity.[1] This kernel captures the geometry of the space of first activations and compactly encodes the inductive biases of that layer.

---

[1]Without loss of generality, we have assumed that $\Phi_1 \in L_2(\mathbb{R}^d)$. If $\Phi_1$ consists of less smooth space of functions than $L_2(\mathbb{R}^d)$, we just define the kernel $k_1$ such that the functions $k_1(\cdot, x)$ for $x \in \mathcal{X}$ are $L_2$-functions, e.g. as $k_1(x, x') = \exp(-(\langle \phi_{1,x}, \phi_{1,x'} \rangle_{\Phi_1})/\sigma^2)$.

By the Moore-Aronszajn theorem (Berlinet & Thomas-Agnan, 2011), the kernel $k_1$ induces a corresponding RKHS $\mathcal{H}_{k_1} = \overline{Span\big(\{k_1(\cdot, x)| \; x \in \mathcal{X}\}\big)}$ which is the closure of the span of so-called canonical feature maps $k_1(\cdot, x)$ of $k_1$. Associated with $\mathcal{H}_{k_1}$, there is an inner product $\langle \cdot, \cdot \rangle_{\mathcal{H}_{k_1}}$ and the *reproducing property* holds, i.e.

$$\forall x \in \mathcal{X}, \; \forall f \in \mathcal{H}_{k_1} \quad \langle f, k_1(\cdot, x)\rangle_{\mathcal{H}_{k_1}} = f(x)$$
$$\forall x, x' \in \mathcal{X} \quad k_1(x, x') = \langle k_1(\cdot, x), k_1(\cdot, x')\rangle_{\mathcal{H}_{k_1}}.$$

Now, in order to construct a second layer of infinite-width, we need to define a distribution over $\Phi_1$ for the weights connecting the first and the second layer. To this end, we use the structure of the induced RKHS $\mathcal{H}_{k_1}$ and the fact that defining a probability distribution over an RKHS can be done in a principled way using the corresponding kernel. The following proposition and lemma summarise our weight initialisation scheme for infinite-width networks.

**Proposition 1.** *Let $x, x' \in \mathbb{R}^d$ be inputs to a network with $l$ infinite-width layers, $k_l$ be the kernel corresponding to the $l$-th layer and $\mathcal{H}_{k_l}$ the induced RKHS. We can extend the network with an $(l+1)$-th layer of infinite width by sampling the weights $w_l$ connecting the $l$-th and $(l+1)$-th layers from a Gaussian process with zero mean function and covariance function*

$$C_l(w, w') = \int k_l(w, w_{l-1}) k_l(w', w_{l-1}) \, d\nu_{l-1}(w_{l-1}) \quad \text{with} \quad w, w' \in \Phi_{l-1}. \tag{3}$$

**Lemma 1** (Network Reparametrisation). *Given the $l$-th layer activations $\Phi_l = \overline{Span\big(\{\phi_{l,x}|x \in \mathcal{X}\}\big)}$, the associated kernel $k_l$ and the induced RKHS $\mathcal{H}_{k_l}$, the mapping*

$$U_l : \Phi_l \to \mathcal{H}_{k_l} \quad \text{with} \quad \phi_{l,x} \mapsto k_l(\cdot, x)$$

*is an isometric isomorphism between $\Phi_l$ and $\mathcal{H}_{k_l}$.*

Proofs of the proposition and lemma are given in the Supplementary Material, but we briefly outline here the main intuitions behind them. In particular, our proposed construction of an infinite-width network proceeds iteratively layer by layer. Having constructed the $l$-th layer, we construct the kernel $k_l$ associated with that layer. This kernel induces an RKHS $\mathcal{H}_{k_l}$. We then perform a reparametrisation of the $l$-th layer activations $\Phi_l$ into $\mathcal{H}_{k_l}$ according to Lemma 1. Having identified $\Phi_l$ with $\mathcal{H}_{k_l}$, we construct the appropriate distribution of the weights connecting layers $l$ and $l + 1$ as the canonical distribution over $\mathcal{H}_{k_l}$. Finally, to construct layer $l + 1$, we sample infinitely many of the connecting weights according to (3).

## 4 INITIALISATION BY PROJECTION

Having established a principled construction approach for infinite-width networks, we now turn our attention to examining the consequences of this construction for standard, finite-width networks. In particular, by reasoning about finite-width networks as Monte Carlo approximations of their infinite-width counterparts, we derive a novel weight initialisation method based on the above weight initialisation scheme for infinite-width networks and examine its structural properties.

In the finite-width case, we can simplify our notation. In particular, given an input to the network $x \in \mathcal{X} \subseteq \mathbb{R}^d$, we denote by $h_l(x)$ the output of layer $l$ and compute its $i$-th entry recursively from its infinite-width counterpart $\phi_{l,x}(w_i^l)$ as

$$[h_l(x)]_i := \hat{\phi}_{l,x}(w_i^{l-1}) = f_l(\langle w_i^{l-1}, \hat{\phi}_{l-1,x}\rangle_{\mathbb{R}^{N_{l-1}}}) = f_l(\langle w_i^{l-1}, h_{l-1}(x)\rangle_{\mathbb{R}^{N_{l-1}}})$$

where $\hat{\cdot}$ refers to the finite-width approximation of the infinite-width quantity. In particular, in the finite-width case, we have $\hat{\Phi}_l \subseteq \mathbb{R}^{N_l}$ with $N_l$ the width of the $l$-th layer and $\hat{\Phi}_l = \overline{Span(\{\hat{\phi}_{l,x}|x \in \mathcal{X}\})}$. Concatenating all the $N_l$ weight vectors that connect layers $l - 1$ and $l$, and lie in $\mathbb{R}^{N_{l-1}}$, we get the weight matrix $W^{l-1} \in \mathbb{R}^{N_l \times N_{l-1}}$.

Most common stochastic initialisation approaches are data- and task-agnostic and draw the weights from a Gaussian distribution with zero mean and appropriately scaled variance. In contrast to that, we obtain a data- and task-dependent approach to initialisation we term *Weight Initialisation with Infinite Networks* (Win-Win) by taking Monte Carlo approximations of the weight initialisation for

infinite-width networks. In particular, we approximate the integrals used for computing the kernels and GP covariance from equations (2) and (3) through sampling, i.e.

$$\hat{k}_l(x, x') = \frac{1}{N_l} \sum_{n=1}^{N_l} f(\langle x, w_n^l \rangle_{\mathbb{R}^{N_{l-1}}}) f(\langle x', w_n^l \rangle_{\mathbb{R}^{N_{l-1}}}) \quad \text{with} \quad \hat{k}_l(\cdot, x)_i = \frac{1}{\sqrt{N_l}} f(\langle x, w_i^l \rangle_{\mathbb{R}^{N_{l-1}}})$$

$$C_l(x, x') \approx \frac{1}{M_l} \sum_{j=1}^{M_l} k_l(x, \xi_j) k_l(x', \xi_j) \approx \frac{1}{M_l} \sum_{j=1}^{M_l} \hat{k}_l(x, \xi_j) \hat{k}_l(x', \xi_j) =: \hat{C}_l(x, x').$$

While the number of samples used in the approximation corresponds to the dimensionality of the feature expansion, i.e. the size of the corresponding finite-width layers, we can also think of these representations as random feature expansions of a particular kernel thus establishing a connection to Rahimi & Recht (2008). In the finite-width case, we have $w_i^l \sim \text{GP}(0, \hat{C}_l)$ which can be written as

$$w_i^l = \sum_{m=1}^{M} \alpha_{m,i} h_l(\xi_{m,i}) \quad \text{with} \quad \alpha_{m,i} \sim \mathcal{N}\left(0, \frac{1}{M}\right)$$

where the coefficients $\alpha_{m,i}$ are mixing weights and $\xi_{m,i}$ are selected training points.

This approach encourages us to think about the interaction between the weights and the hidden layer representations in finite-width networks from a new perspective. In particular, the rows of the weight matrix at layer $l$ are initialised as weighted linear combinations of the activations of some points in $l$-th layer. This facilitates an intuitive interpretation of a neuron $i$ in layer $l+1$ measuring the alignment of the data with the subspace spanned by $\{\xi_{m,i}\}_m$. We propose three different approaches to choosing these subspaces. First, we can choose the points randomly from the training data at every layer; this corresponds to projecting the data onto random subspaces of the training distribution. We denote this approach as Win-Win random. Next, guided by the principle of disentangling the factors of variation in the data, we propose two structured approaches to the subspace selection that incorporate the structure of the data and information about the task at hand directly into the weights already at initialisation. In particular, we disentangle the data manifold by making similar objects increasingly more similar and dissimilar objects increasingly more dissimilar as we move from the lower to the higher layers. For the data and task at hand, we propose two ways to uncover the "disentangled" directions: using k-means clustering and class information, respectively. Pre-training neural networks has a long history of investigation (Hinton et al., 2006; Erhan et al., 2010). Interestingly, the practical methods we propose could be seen as particular kinds of pre-training principally derived from the initialisation of an infinitely-wide neural network. The methods we propose initialise the weights so that the learning dynamics are subtly modified with a bias towards the data manifold.

**K-means-based subspace selection.** For weights connecting layers $l$ and $l+1$, we select the subspaces by clustering the activations of the training data at layer $l$ into $N_{l+1}$ clusters, i.e. one cluster is assigned to every neuron in the subsequent layer. We pick $p$ points from cluster $i$, i.e. $\{\xi_{m,i}^{pos}\}_{m=1}^{p}$, and $n$ points from each of the remaining clusters, i.e. $\{\xi_{m,i}^{j,neg}\}_{m=1}^{n}$ for $j \leq N_{l+1}, j \neq i$, and compute the weight vector $w_i^l$ connecting layer $l$ with neuron $i$ in layer $l+1$ as

$$w_i^l = \sum_{m=1}^{p} \alpha_{m,j} h_l(\xi_{m,i}^{pos}) - \sum_{\substack{j \neq i \\ j \leq N_{l+1}}} \sum_{m=1}^{n} \alpha_{m,j,i} h_l(\xi_{m,i}^{j,neg}). \tag{4}$$

**Class-based subspace selection.** If the task at hand is classification, we can choose the subspaces in a manner that is more aligned with this end goal. Instead of clustering the activations, we use the class labels as cluster assignments. Specifically, we associate every neuron $i$ in layer $l+1$ with a class $c$ and pick $p$ points $\{\xi_{m,i}^{pos}\}_{m=1}^{p}$ from that class. We also pick $\{\xi_{m,i}^{j,neg}\}_{m=1}^{n}$ points from each of the remaining classes $j \neq c$ and compute the weight vector $w_i^l$ as

$$w_i^l = \sum_{m=1}^{p} \alpha_{m,i} h_l(\xi_{m,j}^{pos}) - \sum_{j \neq c} \sum_{m=1}^{n} \alpha_{m,j,i} h_l(\xi_{m,i}^{j,neg}).$$

In both subspace selection methods, points that are from cluster $i$ should activate neuron $i$ more than points from other clusters since we are measuring the alignment to points from that cluster, while substracting the alignment scores of all other clusters. Thus, we are making the points from cluster $i$ more similar to each other and less similar to points from all other clusters.

## 5 EXPERIMENTAL RESULTS

We compared our new initialisation scheme Win-Win to 4 commonly used initialisation schemes, namely *Xavier* (Glorot & Bengio, 2010), *LeCun* (LeCun et al., 2012), *Kaiming* (He et al., 2015) and *SntDefault* (Ioffe & Szegedy, 2015), the default initialiser in Sonnet, using the MNIST dataset (LeCun et al., 1998), the CIFAR-10 dataset (Krizhevsky & Hinton, 2009) and on the Year Prediction MSD dataset (Bertin-Mahieux et al., 2011). For all experiments, we use the validation set for hyperparameter selection and early stopping, and the Adam optimizer (Kingma & Ba, 2014) with the default hyperparameters and a fixed learning rate for the whole of training. Throughout, we used the ReLU non-linearity. Error bars are one standard deviation of the mean and are computed using an unbiased estimate of the variance for 12 different random seeds. We note that we did not use any advanced techniques, such as data augmentation or learning rate scheduling, in order to disentangle the effect of initialisation on network performance from the effects of other training heuristics.

**MNIST.** As a sanity check, we test our method on the benchmark dataset MNIST which encodes handwritten digits from 0 to 9 as grayscale images of size $28 \times 28$ pixels with a train/valid/test split of $50,000/10,000/10,000$ points. We train a 2-layer fully-connected architecture with 800 hidden units. We choose this architecture as they have achieved good performance as reported on http://yann.lecun.com/exdb/mnist/. We test the three different methods for subspace selection in Win-Win – random, class and kmeans. As can be seen from Table 1, random and kmeans-based Win-Win are competitive to all the competing methods. Class-based Win-Win performs worse than other methods which we posit is due to the class information only being informative for initialising the weights connecting to the output layer and not for the weights incoming to the first hidden layer.

Table 1: Final classification error on the MNIST test set.

| Method | Error |
|---|---|
| SntDefault | $1.63 \pm 0.20\ \%$ |
| LeCun | $1.68 \pm 0.24\ \%$ |
| Xavier | $1.85 \pm 0.17\ \%$ |
| Kaiming | $1.63 \pm 0.23\ \%$ |
| Win-Win [random] | $1.67 \pm 0.22\%$ |
| Win-Win [k-means] | $1.58 \pm 0.24\ \%$ |
| Win-Win [class] | $2.17 \pm 0.21\%$ |

In the Supplementary Material, we also study the evolution of activation patterns for the different initialisation methods. In particular, we look at the activations of 100 randomly sampled points from each class immediately after initialisation and during training. For the standard initialisation approaches, the activation pattern across different neurons and classes is rather sparse, see Figures 1-4. At initialisation only very few neurons from the hidden layer contribute to the calculation of the logits. As training progresses, more neurons from the hidden layer get activated, but even after the network has been completely trained, we see that a large number of neurons does not fire for any class. On the other hand, with class-based Win-Win initialisation, we see that neurons assigned to class $c$ fire more prominently for that class than for other classes, while the logits correspond to a uniform prior over the neurons for different classes. As training progresses, we see that this diagonal structure in the hidden layer is roughly preserved without specific constraints.

**CIFAR-10.** This dataset consists of $60,000$ images of size $32 \times 32 \times 3$ pixels with a train/valid/test split of $40\,000/10\,000/10\,000$. As is common practice, we pre-process the images by scaling them to the range $[-1,1]$. For the model, we use a deep convolutional neural network with a fully-connected linear layer as the output layer; details of the architecture are provided in the Supplementary Material. We trained the network for 400 epochs. We only initialised the fully-connected layer according to the different initialisation schemes and the rest initialised with the default initialiser (Ioffe & Szegedy, 2015). We do this in order to demonstrate that Win-Win is beneficial also in situations when it can only be applied to features derived from some transformation of the input data (in this case structured random projections) and not directly to the input data. As can be seen Table 2, the Win-Win initialisation clearly outperforms all other methods by a significant margin.

Table 2: Classification accuracy on the CIFAR10 test set.

| Method | SntDefault | Xavier | LeCun | Kaiming | Win-Win |
|---|---|---|---|---|---|
| Test Accuracy | 85.83% | 86.38% | 86.28% | 86.08% | **87.05 %** |

**Year Prediction MSD dataset.** In addition to the two classification problems summarised above, we assessed the performance of our new initialisation scheme also on a regression task using the Year Prediction MSD dataset (Bertin-Mahieux et al., 2011). In this task, the goal is to estimate the year in which a song was released based on a 90-dimensional input vector of audio features. We split the available data into training, validation and test sets containing $400\,000$, $63\,715$ and $51\,630$ data points, respectively. In a pre-processing step, we then computed the mean and standard deviation of features and labels of the training set, and used these quantities to shift and rescale all three datasets so that the distribution of data points is approximately standard normal.

For comparison, we trained two different, fully-connected architectures with one or two hidden layers followed by a single-unit output layer for a total of 150 epochs with the objective to minimise the mean squared error (MSE). As a natural choice for class-based Win-Win initialisation, we set the number of units in each hidden layer to 89 which corresponds to the number of distinct labels in the training set. As a measure of performance, we use the root mean squared error (RMSE) achieved on the test set (Lakshminarayanan et al., 2017). The experimental results are summarised in Table 3. We can clearly see that Win-Win initialisation outperforms all other schemes in case of a single hidden layer and is comparable to the best performing scheme (SntDefault) for two hidden layers.

Table 3: Regression RMSE values for the Year Prediction MSD dataset. We used SntDefault initialisation for the output layer in case of Win-Win initialisation.

| Method | 1 hidden layer | 2 hidden layers |
|---|---|---|
| SntDefault | $8.9301 \pm 0.0033$ | $\mathbf{8.8798 \pm 0.0072}$ |
| LeCun | $8.9484 \pm 0.0024$ | $8.9414 \pm 0.0086$ |
| Xavier | $8.9376 \pm 0.0044$ | $8.8948 \pm 0.0064$ |
| Kaiming | $8.9448 \pm 0.0039$ | $8.9290 \pm 0.0044$ |
| Win-Win [random/SntDefault] | $\mathbf{8.8902 \pm 0.0022}$ | $\mathbf{8.8693 \pm 0.0060}$ |
| Win-Win [class/SntDefault] | $\mathbf{8.8904 \pm 0.0023}$ | $8.8867 \pm 0.0057$ |

## 6 CONCLUSION

In this paper, we have studied the initialisation requirements of infinite-width networks and have shown that the main challenge in constructing these networks lies in defining the appropriate sampling distributions of the weights. To address this problem, we have presented a novel method for the construction of infinite-width networks that, unlike previous approaches, enables the construction of deep infinite-width networks with arbitrarily many hidden layers. In particular, we have proposed a principled approach to weight initialisation using the theory of reproducing kernel Hilbert spaces. In order to appropriately account for the functional form of the hidden layer activations and to facilitate the construction of arbitrarily many infinite-width layers, we proposed to construct the sampling distributions of the weights at every hidden layer as Gaussian processes with specific covariance kernels that take into account the geometry of the underlying space of activations. To achieve this, we have constructed a hierarchy of kernels that capture the geometry and inductive biases of individual layers in the neural network. Furthermore, using Monte Carlo approximations, we have examined the practical implications of this construction for standard, finite-width networks. In particular, we have derived a novel data- and task-dependent weight initialisation method for this type of network and showcased its competitive performance on three diverse datasets.

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

# 7 SUPPLEMENTARY MATERIAL

## 7.1 PROOFS

For completeness, we restate the proposition and lemma and provide the corresponding proofs.

### 7.1.1 PROPOSITION 1

**Proposition 1.** *Let $x, x' \in \mathbb{R}^d$ be inputs to an network with $l$ infinite-width layers, $k_l$ be the kernel corresponding to the $l$-th layer with $k_l(\cdot, x)$ the corresponding canonical feature maps and $\mathcal{H}_{k_l}$ the induced RKHS. We can extend the network with an $(l+1)$-th layer of infinite width by sampling the weights $w_l$ connecting the $l$-th and $(l+1)$-th layers from a Gaussian process with zero mean function and covariance function*

$$C_l(w, w') = \int k_l(w, w_{l-1}) k_l(w', w_{l-1}) \, d\nu_{l-1}(w_{l-1}) \quad with \quad w, w' \in \Phi_{l-1}$$

*Then $w_l \in \Phi_l$ and $\phi_{l,x}(w_l) = f(\langle w_l, \phi_{l-1,x} \rangle_{\Phi_l})$ is well defined.*

**Proof of Proposition.**
We prove the claim of this proposition by mathematical induction. We first need to establish a correspondence between $\Phi_1$ and $\mathcal{H}_{k_1}$ and construct an *isometric isomorphism* (Coxeter, 1963) between $\Phi_1$ and $\mathcal{H}_{k_1}$. To this end, we invoke Lemma 1 for the first hidden layer. The isometric isomorphism $U_1$ identifies $\Phi_1$ with $\mathcal{H}_{k_1}$ and allows us to replace $\Phi_1$ with the easier to work with $\mathcal{H}_{k_1}$ as the geometry remains unchanged. The transformations involved in the proof are depicted as below.

$$
\begin{array}{ccc}
\Phi_1 & \xleftarrow{\quad U_1 \quad} & \mathcal{H}_{k_1} \\
\downarrow & & \downarrow \\
\phi_{1,x} & \xleftarrow{\quad U_1 \quad} & k_1(\cdot, x)
\end{array}
$$

Using this reparametrization, instead of constructing a distribution for the weights over $\Phi_1$ in order to construct a second layer of infinite width, the problem simplifies to constructing a distribution for the weights over $\mathcal{H}_{k_1}$. In particular, we use the canonical probability distribution over $\mathcal{H}_{k_1}$ to define the sampling distribution for the weights between the first and second infinite-width layer.

Specifically, a probability distribution over $\mathcal{H}_{k_1}$ can be canonically constructed as a Gaussian process with zero mean function and covariance function

$$C_1(w, w') = \int k_1(w, w_0) k_1(w, w_0) \, d\nu_0(w_0) \quad with \quad w, w' \in \Phi_1.$$

This special covariance structure ensures that a GP with this covariance function is a probability distribution over $\mathcal{H}_{k_1}$ (Aronszajn, 1950).

To construct a second hidden layer of infinite-width, we need to sample infinitely many weights $w_1$ connecting the first and second layers from $\nu_1 = GP(0, C_1)$. This construction ensures that the inner product between the first layer representation $\phi_{1,x}$ and the weights connecting the first and second layer is well defined. This, in turn, ensures that the second layer representation $\phi_{2,x}$ given by

$$\phi_{2,k_1(\cdot,x)}(w_1) = f\big( \langle w_1, k_1(\cdot, x) \rangle_{\mathcal{H}_{k_1}} \big)$$

is well defined. We note that $\phi_{2,x}$ is a function over samples from the GP $\nu_1$, i.e. a function of functions over $\mathbb{R}^d$. Associated with the second layer, we define the corresponding kernel as

$$k_2(\phi_{1,x}, \phi_{1,x'}) = \langle \phi_{2,x}, \phi_{2,x'} \rangle_{\Phi_2} \tag{5}$$

with $\Phi_2 = \overline{Span(\{\phi_{2,x} | x \in \mathcal{X}\})}$ and the induced RKHS $\mathcal{H}_{k_2} = \overline{Span(\{k_2(\cdot, \phi_{1,x}) | x \in \mathcal{X}\})}$. The correspondence between these two spaces is given by the isometric isomorphism $U_2 : \Phi_2 \to \mathcal{H}_{k_2}$ with $U_2(\phi_{2,x}) = k_2(\cdot, \phi_{1,x})$ using Lemma 1.

Assuming that we have constructed an infinite-width network with $l$ hidden layers by iteratively repeating the above procedure of defining a kernel corresponding to the hidden layer, we can reparametrise the hidden layer into the induced RKHS and construct the next layer by sampling the connecting weights from the distribution over the induced RKHS. To this end, we first define the kernel $k_l$ corresponding to layer $l$ as

$$k_l(\phi_{l-1,x}, \phi_{l-1,x'}) = \langle \phi_{l,x}, \phi_{l,x'} \rangle_{\Phi_l} = \langle k_l(\cdot, \phi_{l-1,x}), k_l(\cdot, \phi_{l-1,x'}) \rangle_{\mathcal{H}_{k_l}} \quad (6)$$

with $\mathcal{H}_{k_l}$ the induced RKHS. Next, we perform the reparametrisation of that layer into $\mathcal{H}_{k_l}$ using an analogously defined isometric isomorphism $U_l$. We note that both $\phi_{l-1,x}$ and $w_{l-1}$ lie in the same space, i.e. $\Phi_{l-1}$. We construct the distribution for the weights connecting the $l$-th and $(l+1)$-th layer as a GP $\nu_l$ with zero mean and covariance function given by

$$C_l(w, w') = \int k_l(w, w_{l-1}) k_l(w', w_{l-1}) \, d\nu_{l-1}(w_{l-1}) \quad \text{with} \quad w, w' \in \Phi_{l-1}$$

and sample infinitely many weights from that distribution. Finally, we propagate the data along the constructed weights to add another hidden layer of infinite-width. At layer $l + 1$, this yields

$$\phi_{l+1,x}(w_l) = f(\langle w_l, k_l(\cdot, \phi_{l,x}) \rangle_{\mathcal{H}_{k_l}}) \quad \text{with} \quad w_l \sim \nu_l(w_l) = \mathrm{GP}(0, C_l).$$

The last step completes the addition of layer $l + 1$. $\qquad\square$

### 7.1.2 LEMMA 1

**Lemma 1** (Network Reparametrisation). *Let $\Phi_l$ denote the representational space of layer $l$, i.e.*

$$\Phi_l = \overline{Span(\{\phi_{l,x} | x \in \mathcal{X}\})},$$

*and $k_l$ be the associated kernel at layer $l$ and $\mathcal{H}_{k_l}$ be the corresponding RKHS to kernel $k_l$. The mapping*

$$U_l : \Phi_l \to \mathcal{H}_{k_l} \quad \text{with} \quad \phi_{l,x} \mapsto k_l(\cdot, x)$$

*is an isometric isomorphism between $\Phi_l$ and $\mathcal{H}_{k_l}$.*

**Proof of Lemma.** Without loss of generality, we restrict ourselves to proving the lemma for the first hidden layer of infinite-width. For all subsequent layers, the proof proceeds analogously.

We want to prove that

$$U_1 : \Phi_1 \to \mathcal{H}_{k_1} \quad \text{with} \quad \phi_{1,x} \mapsto k_1(\cdot, x)$$

is an isometric isomorphism. In particular, we need to prove that $U_1$ is a bijective map that satisfies

$$\langle \phi_{1,x}, \phi_{1,x'} \rangle_{\Phi_1} = \langle U_1(\phi_{1,x}), U_1(\phi_{1,x'}) \rangle_{\mathcal{H}_{k_1}}.$$

Using the reproducing property and the definition of $k_1$ (2), we find

$$\langle U_1(\phi_{1,x}), U_1(\phi_{1,x'}) \rangle_{\mathcal{H}_{k_1}} = \langle k_1(\cdot, x), k_1(\cdot, x') \rangle_{\mathcal{H}_{k_1}} = k_1(x, x') = \langle \phi_{1,x}, \phi_{1,x'} \rangle_{\Phi_1}$$

which shows that $U_1$ is an isometric isomorphism between $\Phi_1$ and $\mathcal{H}_{k_1}$. Bijectivity can be directly derived from the definition. The construction of $U_1$ can intuitively be understood as a reparametrisation of the first hidden layer representations $\phi_{1,x}$ into the canonical feature maps $k_1(\cdot, x)$. Furthermore, the $\Phi_l$ inner product between the first hidden layer representations can now be calculated as the inner product between the corresponding canonical feature mappings in the RKHS. This result guarantees that the geometry will be unchanged as the spaces $\Phi_l$ and $\mathcal{H}_{k_l}$ are identified by an isometric isomorphism. $\qquad\square$

### 7.2 ACTIVATION PATTERNS

The following heatmaps represent activation patterns of 100 randomly sampled points from each class at initialisation and at epoch 20 and 400 during training. The left column encodes the activations of the first hidden layer, while the right column encodes the output layer. To all activations the softmax function has been applied. The $x$-axis encodes the class of a datapoint, while th $y$-axis encodes the

neurons. For the first hidden layer where there are more than 10 neurons, we construct consecutive buckets of neurons of one tenth of the layer width and average over these.

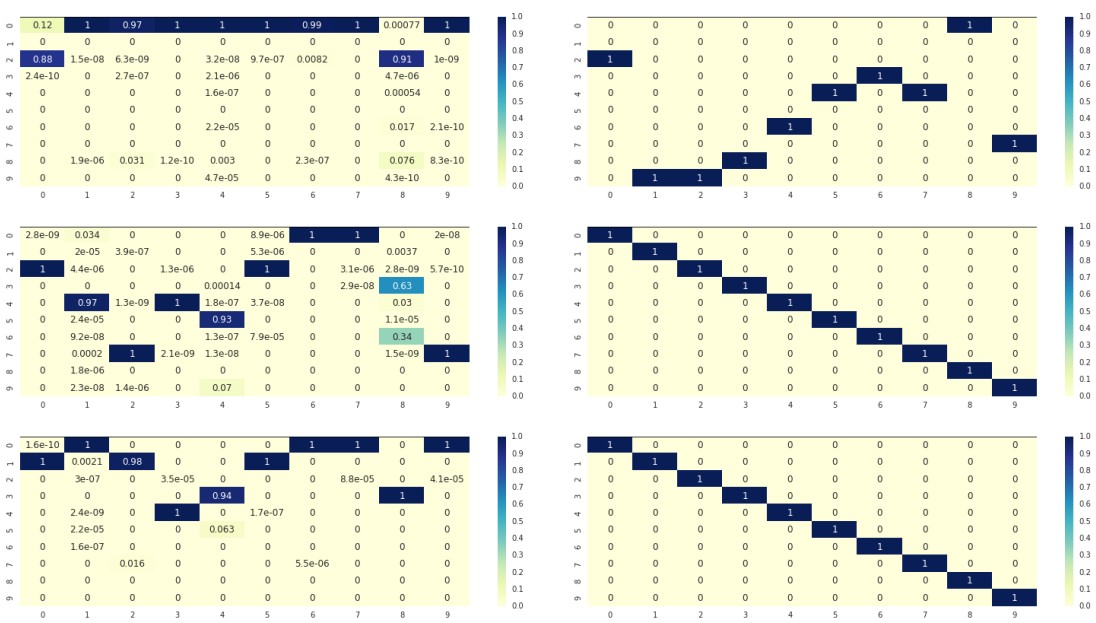

Figure 1: Softmaxed activations of the hidden and output layer for LeCun initialisation at epochs 0, 20 and 400 for a network with 800 hidden units.

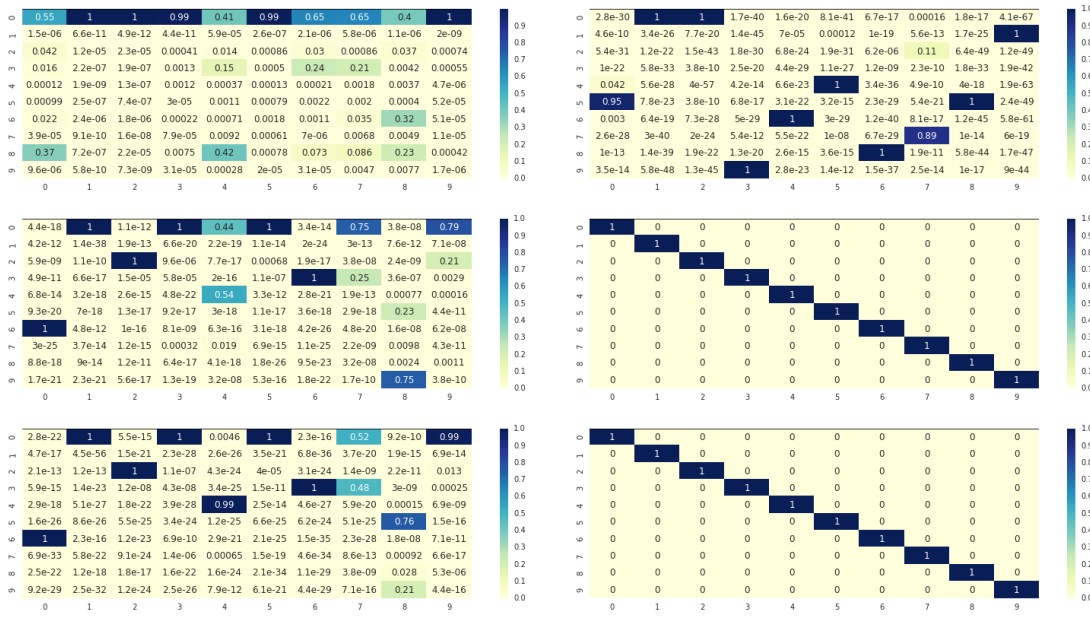

Figure 2: Softmaxed activations of the hidden and output layer for Xavier initialisation at epochs 0, 20 and 400 for a network with 800 hidden units.

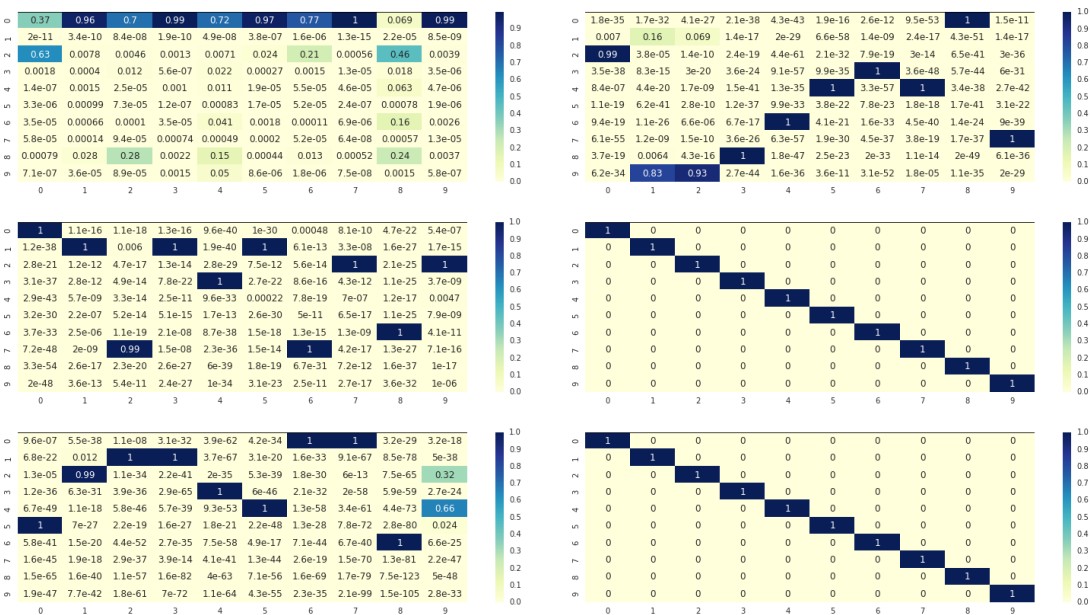

Figure 3: Softmaxed activations of the hidden and output layer for Kaiming initialisation at epochs 0, 20 and 400 for a network with 800 hidden units.

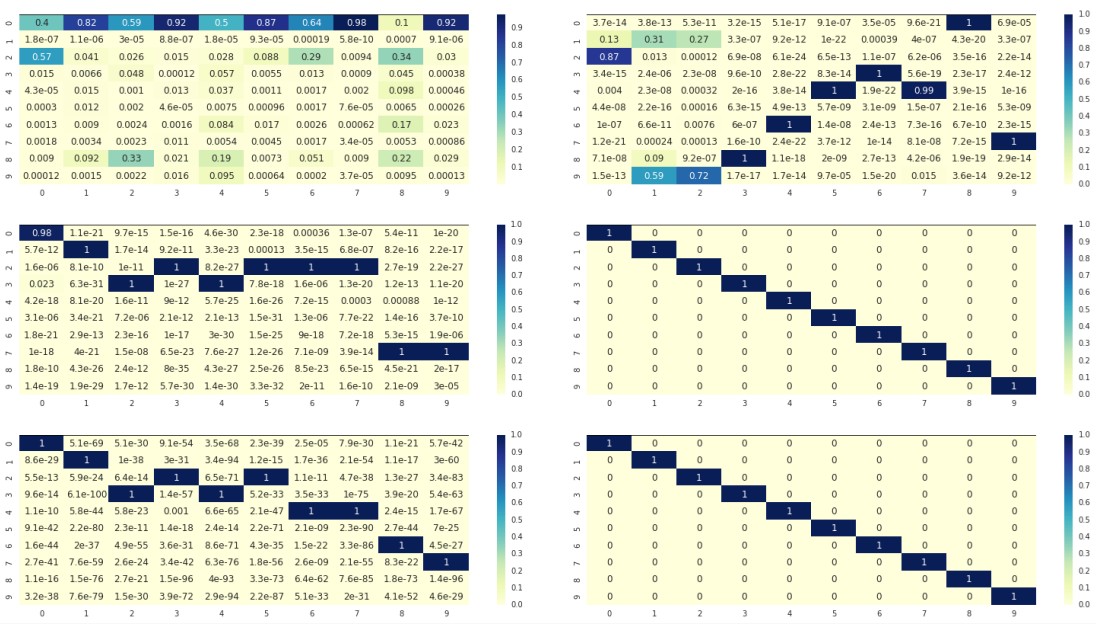

Figure 4: Softmaxed activations of the hidden and output layer for SntDefault initialisation at epochs 0, 20 and 400 for a network with 800 hidden units.

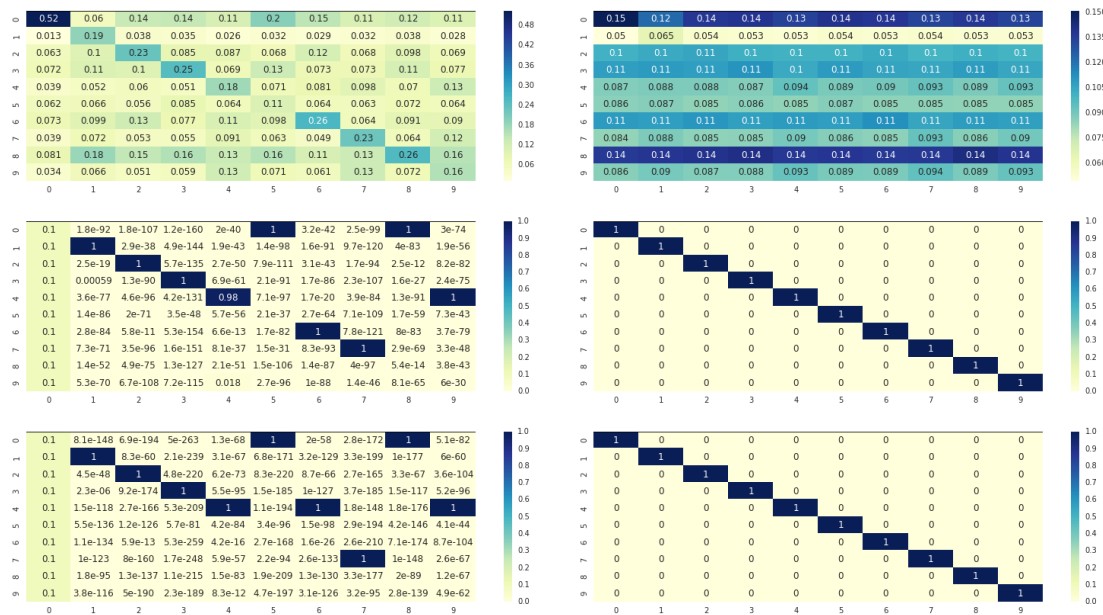

Figure 5: Softmaxed activations of the hidden and output layer for Win-Win class initialisation at epochs 0, 20 and 400 for a network with 800 hidden units.

## 7.3 CONVOLUTIONAL ARCHITECTURE FOR CIFAR-10

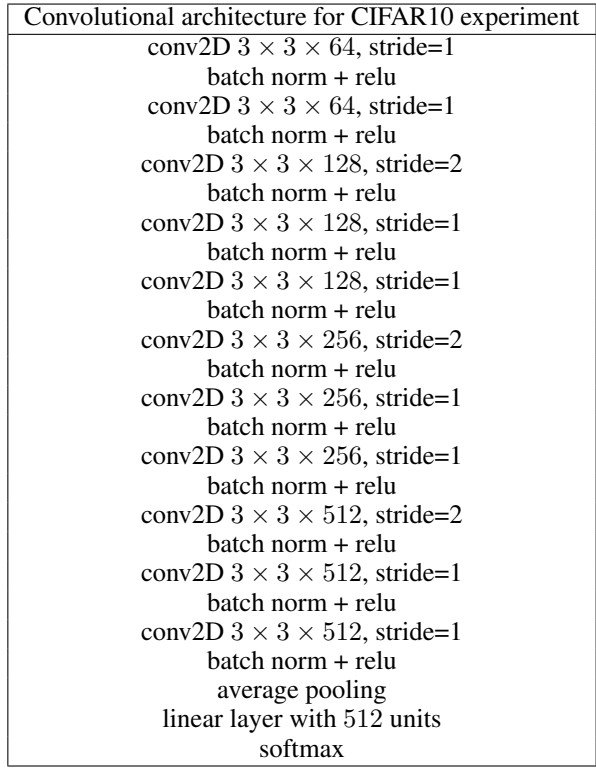

| Convolutional architecture for CIFAR10 experiment |
|---|
| conv2D $3 \times 3 \times 64$, stride=1 |
| batch norm + relu |
| conv2D $3 \times 3 \times 64$, stride=1 |
| batch norm + relu |
| conv2D $3 \times 3 \times 128$, stride=2 |
| batch norm + relu |
| conv2D $3 \times 3 \times 128$, stride=1 |
| batch norm + relu |
| conv2D $3 \times 3 \times 128$, stride=1 |
| batch norm + relu |
| conv2D $3 \times 3 \times 256$, stride=2 |
| batch norm + relu |
| conv2D $3 \times 3 \times 256$, stride=1 |
| batch norm + relu |
| conv2D $3 \times 3 \times 256$, stride=1 |
| batch norm + relu |
| conv2D $3 \times 3 \times 512$, stride=2 |
| batch norm + relu |
| conv2D $3 \times 3 \times 512$, stride=1 |
| batch norm + relu |
| conv2D $3 \times 3 \times 512$, stride=1 |
| batch norm + relu |
| average pooling |
| linear layer with 512 units |
| softmax |

