# OpenReview forum: "Infinitely Deep Infinite-Width Networks"
_ICLR.cc/2019/Conference_

### Official Review · AnonReviewer2 · 2018-10-31
**a weight initialization approach to enable infinitely deep and infinite-width networks, experimental results on small datasets**

**Rating:** 6
**Confidence:** 4

**Review:**

Pros:

This paper uses kernel mappings between any two layers for weight initialisation. Using the representer theorem, a proper distribution for weights is constructed in H_{k_i} instead of being learned by \phi_i, and then is formulated as a GP.

Cons:

However, there are some key issues.
1. The so-called “infinite width” is just yielded by kernels in RKHS for weight initialization. For practical implementation, the authors use this scheme with random Fourier features to construct finite-width network. A key issue is that how to guarantee that the approximated weights are still in the same space? For example, weights can be in RKHS, but their approximation might be not in RKHS. See in [S1] for details.

[S1] Generalization Properties of Learning with Random Features, NIPS 2017.

2. Experimental part is not very convincing. First, the authors just compare different initialization schemes. The used architectures are simple and not representative. Second, the overall performance is not satisfactory, and the compared classification datasets are quite small. Overall, the experimental results are inadequate and unconvincing.

Summary:
The paper attempts to proposal a weight initialization scheme to enable infinite deep infinite-width networks. However, there are some key issues not address such as whether the approximated weights are still in the same space and the limited experimental results.

Response to rebuttal:
The authors have addressed my question about the weights being still in the same RKHS. I still think the motivation and experiments are not very satisfactory.

Therefore the paper is very borderline. However, I would like to bump my rating a bit higher.

---

> ### Author Response · Authors · 2018-11-26
> **Theoretical contribution validated by experimental results**
>
> We thank the reviewer for their time and comments.
>
> In our proposed method, we define a kernel at the level of a layer and construct the distribution over the weights based on that kernel. In particular, our method does not construct kernel mappings between layers and as such does not use these for weight initialisation. Furthermore, the distribution constructed over the RKHS H_{k_i} is not a result of the representer theorem as no objective is being minimised with respect to any training data. Specifically, this distribution is derived by studying the structure of the induced RKHS and is a known result from RKHS theory. Given the constructed weight distribution, this particular form of the weights arises due to the fact that the covariance function of the GP is a convolution of kernels. This can be easily verified by computing the covariance between the weights. Note that although the form of the weights does resemble the result from the representer theorem, it is not based on it, but follows from results on RKHS distributions as discussed above.
>
> Before addressing the specific issues identified by the reviewer, we would first like to draw the attention of the reviewer to the main contribution of this paper (Proposition 1 in our submission), namely a method for constructing infinite-width networks with arbitrarily many layers. To the best of our knowledge, this is the first method that allows us to go beyond just two layers of infinite width. As single-layer infinite-width networks have historically played an important role in helping us acquire a deeper understanding of standard, finite-width networks, being able to construct deep infinite-width networks should help us gain a better understanding of deep, finite-width networks.
>
> Although the main contribution of this paper is of theoretical nature, we also discuss its immediate practical implications. In particular, we can transfer our findings from the infinite-width to the finite-width case. Specifically, we show how our proposed infinite-width construction approach gives rise to a novel weight initialisation scheme for finite-width networks that we term Win-Win. Furthermore, we conducted experiments with Win-Win on benchmark datasets across two different task domains, classification and regression, thus showcasing that our theoretical contribution has direct and practical relevance.
>
> We now address the issues raised by the reviewer:
> The infinite width of the layers is given by the fact that there are infinitely many weights connecting each pair of layers. In particular, the infinite width is not given by any kernel as we define our kernels at the level of individual layers that are of infinite width to begin with. Further, we would like to draw the attention of the reviewer to the fact that, after proposing a construction approach for infinite-width networks with arbitrarily many layers, we examine the implications of this construction for finite-width networks. In particular, the practical implications discussed are motivated by the idea of transferring the findings from infinite-width networks to their finite-width counterparts and thus hopefully enabling a better understanding of finite-width networks in general. Note that this is not done with the idea of practically implementing the proposed infinite-width construction approach.
> Furthermore, we note that we use Monte Carlo sampling to approximate the integrals derived in the infinite-width case. As we discuss in the paper, this can be viewed as a random features expansion of the kernels. Note, however, that in general this will not be a random Fourier features expansion because the underlying kernels are not guaranteed to be shift invariant.
>
> With regard to the comment about ensuring that the “approximated weights are still in the same space”, we would like to point out that, in the finite-width case, this is automatically ensured as the weights are linear combinations of the activations. In particular, in the finite-width case, the required spaces are all subspaces of R^d , where d is the width of the respective layer. Thus, by examining the dimensionality of the weight vectors, it is straightforward to verify that the weights are indeed in the appropriate space. We understand this potential source of confusion and will provide further clarification in the paper.

---

> > ### Author Response · Authors · 2018-11-26
> > **Continued: Theoretical contribution validated by experimental results**
> >
> > Concerning the experimental results -- Note that the main contribution of this paper is of theoretical nature (Proposition 1 in our submission). Furthermore, we extensively discuss the practical implications of the developed theory and conduct experiments on benchmark datasets in two different domains, classification and regression, to demonstrate the direct and practical relevance of our theory. Given that the main motivation of this paper is expanding our understanding of neural networks, the discussed practical implications and experimental results should be seen as highlighting the practical relevance of the proposed theory and not as an attempt at developing a new training heuristic for achieving state-of-the-art performance.
> >
> > With regard to the comment about unconvincing results, we think that this should be put into perspective with the goal of the paper and the architectures considered. In order to adequately validate the developed theory, we need to disentangle the effect of initialisation on network performance from the effects of other training heuristics. Thus, we purposefully did not use any advanced techniques, such as data augmentation, learning rate scheduling or architecture search, all of which are commonly used to improve state-of-the-art performance.
> >
> > Thus, to appropriately validate our theory, we tested Win-Win against commonly used initialisation methods on three benchmarks datasets in classification and regression, thus showing that the performance of Win-Win initialisation is not tied to any particular dataset or task. On all datasets considered in the paper, we have demonstrated that Win-Win either matches or exceeds the performance of the competing approaches for the considered architectures. In particular, for classification on MNIST, we chose the architectures that had the best performance without using advanced preprocessing and training heuristics as recorded on http://yann.lecun.com/exdb/mnist/. For CIFAR10, we used an architecture that achieves state-of-the-art performance and applied the different initialisation approaches in the last fully-connected layer. Note that we didn’t use any advanced training heuristics here, but just made use of the underlying architecture.
> >
> > We hope that our reply to the reviewer’s comment about experimental results explains our main motivation for choosing the tasks, architectures and datasets for the experimental studies presented in the paper, and thereby also addresses the reviewer’s final comment on our results being “ inadequate and unconvincing”. If you have any further comments, suggestions or questions, we would appreciate them.
> >
> > Given the above explanations, particularly our responses to the raised issues, we would kindly like to ask you to consider raising the rating of this paper. Thank you!

---

### Official Review · AnonReviewer3 · 2018-11-03
**Systematic theoretical work with some supporting experiments**

**Rating:** 6
**Confidence:** 2

**Review:**

Summary:
I'm not very familiar with the work this is building off of, but my summary is as follows:
The authors look at the problem of defining multilayer infinite width neural networks. The main challenge is that the weights (which are in some sense now a function) must be appropriately sampled to ensure that norms don't explode.

This has only been done for two layers before, and the authors derive how to do this for more than two hidden layers using RKHSs. This initialization is called Win-Win, and is compared to different initializations on a few different datasets.

Clarity: This paper is quite technical and hard to follow without knowledge of the prior work. I think the authors could have been a little clearer on some of the challenges. E.g. instead of talking about the weights needing to be "in the same function space", it would be helpful to remphasise the norm issue.

Comments:
I'm not sure about the high level motivation for developing networks of this kind. In particular, none of the performance numbers are near state of the art (not necessary for developing new promising methods!) but I don't know exactly what the initialization buys in this setting.

Also, it would have been nice to see whether the Win-Win initialization is only useful for larger width networks compared to smaller width i.e. do other initialization schemes work better in this latter setting?

The derivations are interesting though, so I still recommend accept.

---

> ### Author Response · Authors · 2018-11-12
> **Novel theoretical contribution with practical consequences supported by experiments**
>
> We thank the reviewer for their time and comments.
>
> In the case of infinite-width networks, both the activations and weight “vectors” (connecting one layer with one neuron in the subsequent layer) are infinite-dimensional “vectors”, i.e. they are representable by functions. In order to be able to compute the inner product between activations and weights, these objects need to be in the same function space. If they are not in the same function space, then we cannot compute inner products between these quantities and therefore cannot define infinite-width networks correctly. If we assume that the activations are sufficiently well-behaved then the function space that they span will be a subset of, for example, L_{2}. In that case, we can characterise these functions in terms of their norms.
>
> With regard to the comment on the norm issue, in general we do not know how well-behaved activations are and are therefore not able to characterize the space of activations in terms of norms. Thus, in general, we have no way of quantifying the space of activations in terms of norms and cannot further address this issue apart from emphasizing that the activations and weights need to be in the same function space.
>
> The main contribution of this paper is theoretical, namely a method to construct infinite-width networks with arbitrarily many layers (Proposition 1 in our submission). To the best of our knowledge, this is the first method that allows us to go beyond just two layers of infinite width. Historically, single-layer infinite-width networks have played an important role in helping us acquire a deeper understanding of standard, finite-width networks. Being able to construct deep infinite-width networks should help us gain a better understanding of deep, finite-width networks that have transformed the field dramatically over the last decade. This summarises our main motivation for developing a construction approach for deep infinite-width networks.
>
> Apart from purely theoretical merit, our theoretical contribution also has immediate practical implications. In particular, we can transfer some of our findings from the infinite-width case to finite-width networks. Specifically, we show how our proposed construction approach gives rise to a novel weight initialisation scheme for finite-width networks that we term Win-Win.
>
> Due to the theoretical nature and mathematical complexity involved, we agree that this paper is rather technical. We therefore tried to ease the reader into the challenges of defining deep infinite-width networks (see Section 3.1). We thank the reviewer for the comment about clarity and welcome any further comments on what we could add to the text to make this topic more approachable.
>
> Although the main contribution of this paper is theoretical in nature, we also discuss some practical implications of the developed theory. Note that the main motivation of this paper is expanding our understanding of neural networks. Thus, the practical implications discussed should be seen as highlighting the practical relevance of the proposed theory and not as an attempt at developing a new training heuristic for achieving state-of-the-art performance.
>
> To highlight the implications of the developed theory, we conducted experiments on benchmark datasets across different task domains. In particular, the experimental results are there to demonstrate that our theoretical contribution has direct and practical relevance, and to validate the theory developed in the paper.
>
> Furthermore, in order to adequately validate the developed theory, we needed to disentangle the effect of initialisation on network performance from the effects of other training heuristics. To this end, we purposefully did not use any advanced techniques, such as data augmentation or learning rate scheduling, all of which are commonly used to improve state-of-the-art performance. In the experimental setup, we tested Win-Win against commonly used initialisation methods on three benchmark datasets in classification and regression, thus showing that the advantages of Win-Win are not tied to one particular dataset or even one particular task.
>
> If you have any further comments, suggestions or questions, we would appreciate them. Given the above explanations, we would also kindly like to ask you to consider raising the rating of this paper. Thank you!

---

### Official Review · AnonReviewer1 · 2018-11-09
**The idea proposed in the paper is interesting but the paper appears quite incomplete**

**Rating:** 5
**Confidence:** 4

**Review:**

In this paper, the authors propose deep neural networks of infinite width. The primary challenge in such networks is defining a distribution over the weights connecting two layers of infinite width. The authors tackle this by using Gaussian Processes for these distributions with the covariance functions defined in a canonical manner. Inspired by these networks, the authors propose weight initialization schemes for finite width networks.

While the idea proposed in the paper is interesting, the paper appears quite incomplete. In particular, the authors do not mention a single example of a kernel that can be constructed using the process outlined in Section 3. Furthermore, the only application of these infinitely wide networks proposed in this paper is for initialization of the weights of finite width networks. It will perhaps be more interesting if the authors can use the kernels obtained for supervised learning tasks using kernel machines (as done in Cho & Saul 2009) or as the covariance function of a Gaussian process (as done in Wilson et al. 2014).

Moreover, the experiments conducted on finite width networks are not enough to justify the utility of this initialization scheme. It will be useful if the authors can test the performance of the state-of-the-art networks for CIFAR-10/100 and ImageNet, where the weights of the last fully connected layer have been sampled from different distributions. An extension to initialization for convolutional layers will further strengthen the paper.

---

> ### Author Response · Authors · 2018-11-27
> **Novel theoretical contribution validated by experimental results**
>
> We thank the reviewer for their time and comments.
>
> Concerning examples of kernels that can be constructed using our proposed approach in Section 3, the resulting kernels have been derived for some specific nonlinearities (e.g. ReLU nonlinearities by Cho & Saul 2009). Given that our proposed approach is agnostic to the choice of non-linearity, due to the iterative nature of the construction, the derived kernels will in general not be available in closed form and are thus not particularly suited for use within e.g. an SVM. However,  one can examine the structural properties of these kernels. We will clarify this point in the paper.
>
> Unlike Cho & Saul 2009 and Wilson et al 2016 who strive to derive kernels that mimic the computations in deep networks, this paper is motivated by the desire to improve our understanding of neural networks. In particular, we are motivated by the fact that single-layer infinite-width networks have played an important role in helping us acquire a better understanding of single-layer standard, finite-width networks. Until now the construction of infinite-width networks has been limited to at most two layers, while on the other hand we currently use deep finite-width networks. Our paper is an effort towards bridging this gap by enabling the construction of deep infinite-width networks that can then be used for analysing deep finite-width networks. This should help us gain a better understanding of deep, finite-width networks that have transformed the field dramatically over the last decade. Please also note that in this paper kernels are used as tools that enable us to more easily reason about the function spaces induced by the activations of an infinite-width network and are not the main focus of the paper. In particular, the main motivation of our paper is not the derivation of kernels that mimic computations in infinite-width neural networks, but the furthering our understanding of neural networks.
>
> Concerning the experimental results -- The main contribution of this paper is of theoretical nature (see Proposition 1 in our submission). However, we also extensively discuss the practical implication of this results and support the direct practical relevance of our theory with experiments on benchmark datasets in two different domains, classification and regression. Given the motivation for our paper, the experimental results should be seen as showing the practical relevance of the proposed theory and not as an attempt at developing a new training heuristic for achieving state-of-the-art performance.
>
> To adequately validate our proposed theoretical contribution, we need to disentangle the effect of initialisation on network performance from the effects of other training heuristics. Thus, we purposefully did not use any advanced techniques, such as data augmentation, learning rate scheduling or architecture search, all of which are commonly used to improve state-of-the-art performance.
>
> In particular, to appropriately validate our theory, we tested Win-Win against commonly used initialisation methods on three benchmarks datasets in classification and regression, thus showing that the performance of Win-Win initialisation is not tied to any particular dataset or task. On all datasets considered in the paper, we have demonstrated that Win-Win either matches or exceeds the performance of the competing approaches for the considered architectures. In particular, for classification on MNIST, we chose the architectures that had the best performance without using advanced preprocessing and training heuristics as recorded on http://yann.lecun.com/exdb/mnist/. For CIFAR10, we used an architecture that achieves state-of-the-art performance and applied the different initialisation approaches in the last fully-connected layer. Note that we didn’t use any advanced training heuristics here, but just made use of the underlying architecture.
>
> With respect to the reviewer’s comment about our experimental setup, we hope that our reply explains our main motivation for choosing the tasks, architectures and datasets for the experimental studies presented in the paper.
>
> Following the reviewer’s suggestion, we have now also implemented Win-Win for convolutional layers and are currently running experiments. We will update this thread and the paper with the results as soon as they are available.
>
> If you have any further comments, suggestions or questions, we would appreciate them.
> Given the above explanations, particularly our responses to the raised issues, we would kindly like to ask you to consider raising the rating of this paper. Thank you!

---

### Comment · Area_Chair1 · 2018-11-30
**Has an infinite deep infinite-width network being used?**

Dear authors,

As stated in your abstract: "we propose a principled approach to weight initialisation that correctly accounts for the functional nature of the hidden layer activations and facilitates the CONSTRUCTION of arbitrarily many infinite-width layers, thus enabling the CONSTRUCTION of arbitrarily deep infinite-width networks," but all I could find in your paper are networks with fixed depths and widths. If you can show your method helps CONSTRUCT a network whose depth and width are learned from the data and could in theory go to infinite, then I could buy your claim, but at this moment, I can hardly see the reason for claiming "Infinitely Deep Infinite-Width Networks" in the title and elsewhere. Could you please elaborate on this point?

Thanks,
AC

---

### Public Comment · (anonymous) · 2018-11-30
**Clarifications regarding Proposition 1**

I have several confusions regarding Proposition 1. I will try to informally describe the proof in Proposition 1. Kindly clarify if my understanding is correct or not.

Firstly regarding the statement of the proof: You are trying to construct a distribution over the weights connecting two (infinitely wide) hidden layers that will ensure that the inner product between the activations and the weights is well defined. Is this correct?

Now regarding the proof itself: Firstly, you define a distribution over the weights connecting the first and second hidden layer. Since the first hidden layer has infinite width, any weight vector connecting the first hidden layer will be infinite dimensional. We can think of this weight vector as a function of the weights connecting the visible layer and the first hidden layer. Is this correct? To be more specific, a weight vector connecting the first infinite hidden layer to a single unit of the second hidden layer is treated as a function of the weights between the visible and hidden layer?

Assuming the above is indeed correct, you now need to define a distribution over these weights. In particular, you assume that these weights are sampled from a Gaussian Process with 0 mean and covariance function C_1. Now you claim that this choice of C_1 ensures that resultant distribution is a distribution over the RKHS induced by the first hidden layer. This is confusing for me since I am not aware of this result. Can you please point me to the exact result in (Aronszajn) where this result is mentioned?

I think that the entire theoretical aspect of the paper relies on this result (presumably from Aronszajn). Hence, it is surprising that this result is not mentioned in the "Related Works" section or anywhere else in the main paper. With this result in mind, the rest of the paper becomes easy to digest.

---

### Meta-Review · Area_Chair1 · 2018-12-11
**Solid theoretical analysis but unconvincing experiments and limited potential impacts**

**Confidence:** 4
**Recommendation:** Reject

**Metareview:**

The paper studies how to construct infinitely deep infinite-width networks from a theoretical point of view, and uses the results of its theoretical analysis to design a weight initialization scheme for finite-width networks. While the idea is interesting and the paper may contain novel theoretical contributions, the experimental results are weak, as pointed out by all three reviewers from several different perspectives. In particular, it seems that the presented theoretical analysis is useful mainly for weight initialization and hence has limited potential impacts. In addition, the authors have responded to neither the AC's question, nor a detailed anonymous comment that challenges the value of Proposition 1 given the previous work by Aronszajn.